

# Elucidating the regulatory role of long non-coding RNAs in drought stress response during seed germination in leaf mustard

Jinxing Wei[1,*], Haibo Li[2,*], Xiaoer Huang[1], Yongguo Zhao[1], Lejun Ouyang[1], Mingken Wei[1], Chun Wang[1], Junxia Wang[3] and Guangyuan Lu[1]

[1] Guangdong University of Petrochemical Technology, Maoming, China
[2] Shaoguan University, Shaoguan, China
[3] South China Agricultural University, Guangzhou, China
[*] These authors contributed equally to this work.

Corresponding authors
Junxia Wang,
junxiawang@scau.edu.cn
Guangyuan Lu, luwiz@163.com

## ABSTRACT

Leaf mustard (*Brassica juncea* L. Czern & Coss), an important vegetable crop, experiences pronounced adversity due to seasonal drought stress, particularly at the seed germination stage. Although there is partial comprehension of drought-responsive genes, the role of long non-coding RNAs (lncRNAs) in adjusting mustard's drought stress response is largely unexplored. In this study, we showed that the drought-tolerant cultivar 'Weiliang' manifested a markedly lower base water potential ($-1.073$ MPa *vs* $-0.437$ MPa) and higher germination percentage (41.2% *vs* 0%) than the drought-susceptible cultivar 'Shuidong' under drought conditions. High throughput RNA sequencing techniques revealed a significant repertoire of lncRNAs from both cultivars during germination under drought stress, resulting in the identification of 2,087 differentially expressed lncRNAs (DELs) and their correspondingly linked 12,433 target genes. It was noted that 84 genes targeted by DEL exhibited enrichment in the photosynthesis pathway. Gene network construction showed that MSTRG.150397, a regulatory lncRNA, was inferred to potentially modulate key photosynthetic genes (*Psb27*, *PetC*, *PetH*, and *PsbW*), whilst MSTRG.107159 was indicated as an inhibitory regulator of six drought-responsive *PIP* genes. Further, weighted gene co-expression network analysis (WGCNA) corroborated the involvement of light intensity and stress response genes targeted by the identified DELs. The precision and regulatory impact of lncRNA were verified through qPCR. This study extends our knowledge of the regulatory mechanisms governing drought stress responses in mustard, which will help strategies to augment drought tolerance in this crop.

## INTRODUCTION

Leaf mustard (*Brassica juncea* L. Czern & Coss) is an important vegetable crop cultivated in various regions including China (*Thakur et al., 2020*). Mustard exhibits a high susceptibility to abiotic stresses, which can have a detrimental impact on its yield (*Saha et al., 2016*).

For instance, it is susceptible to seasonal drought during germination, which can affect subsequent seedling emergence and seedling survival (*Chauhan et al., 2007*; *Aneja et al., 2015*; *Saini et al., 2019*; *Alamri et al., 2020*).

Plants employ a plethora of mechanisms to alleviate drought stress during seed germination and subsequent development, including the upregulation of photosynthesis and its associated pathways, which are crucial components of the plant's response to abiotic stress (*Meng, Wen & Zhang, 2022*; *Zhou et al., 2023*). Light acts as a fundamental environmental cue that initiates the morphogenesis of plant seedlings. RNA-seq technology serves as a powerful tool for gaining insights into how plants modulate their physiological pathways in response to environmental signals. For example, an RNA-seq analysis of soybean seed germination at 18 h revealed a substantial induction of the photosynthesis pathway, underscoring its significance in the early phases of seed germination (*Hu et al., 2021*). The adaptive response of photosynthesis to drought stress primarily aims to uphold carbon assimilation and tissue metabolic activities, enabling cells to reestablish equilibrium and endure challenging environmental conditions (*Chaves, Flexas & Pinheiro, 2009*). Nonetheless, the understanding of photosynthesis-related mechanisms in mustard's response to drought stress during seed germination is lacking.

While gene families such as myeloblastosis (MYB), WUSCHEL-related homeobox (WOX), superoxide dismutase (SOD), and ascorbate peroxidase (APX) have been shown to significantly impact the growth, development, and responses to abiotic stress in mustard (*Verma, Lakhanpal & Singh, 2019*; *Verma, Upadhyay & Singh, 2022*; *Xie et al., 2023a*; *Xie et al., 2023b*; *Yang et al., 2023*), the *Aquaporin* gene family and Late Embryogenesis Abundant (LEA) gene family, crucial for regulating seed germination and stress response, have not been studied in mustard, limiting our understanding of drought-tolerant breeding in this crop. Aquaporins, as transmembrane proteins forming water channels, play a vital role in water absorption during seed germination (*Chen, Fessehaie & Arora, 2013*; *Chaumont & Tyerman, 2014*; *Hoai et al., 2020*), closely associated with drought stress and essential for regulating germination timing, necessary base water potential, and maintaining cell osmotic pressure (*Singh et al., 2022*). Similarly, LEA proteins, which are low molecular weight proteins synthesized in the late stages of seed development, are crucial for plant adaptation to environmental stresses, particularly drought stress (*Shao, Liang & Shao, 2005*; *Liang et al., 2016*). Overall, research on the mechanisms of drought tolerance in mustard remains limited, particularly regarding the transcriptional regulatory mechanisms during the seed germination phase under drought stress.

The gene families and individual genes implicated in stress response mechanisms are subject to diverse regulatory processes in plants, including control by non-coding RNAs and epigenetic modifications (*Bashir et al., 2019*). Long non-coding RNAs (lncRNAs) are particularly noteworthy for their critical roles in various biological processes, such as development and stress responses (*Yu et al., 2019*). These lncRNAs, characterized by RNA molecules exceeding 200 nucleotides that do not encode proteins, contribute to abiotic stress responses by modulating mRNA expression levels (*Qin et al., 2017*). While some lncRNAs responsive to drought stress have been identified in mustard using limited sample sizes (*Bhatia et al., 2020*), a comprehensive regulatory framework is yet to be

elucidated. Research in rapeseed has shown that specific lncRNAs regulate drought stress by modulating plant hormone signaling pathways (*Tan et al., 2020*). Similarly, in Chinese cabbage (*B. rapa* ssp. chinensis), it was observed that lncRNAs and miRNAs regulate heat stress responses through pathways associated with plant hormones (*Wang et al., 2019*). Using publicly available RNA-seq data, exploring lncRNAs and their regulatory networks in plants has become a feasible strategy (*Chen, Zhong & Qi, 2021*; *Kumar et al., 2023*; *Xie et al., 2023a*; *Xie et al., 2023b*). For instance, under drought stress conditions, *Chen et al. (2021)* identified a set of upregulated lncRNAs involved in regulating genes related to carbon fixation, chlorine metabolism, and fatty acid synthesis.

In our previous study, we employed RNA-seq analysis to probe into the mechanisms governing drought stress during mustard seed germination under both control and drought-stressed conditions at various time points (*Wei et al., 2023*). While this investigation enabled the identification of numerous protein-coding genes, the exploration of non-coding genes, such as lncRNAs, was lacking. Building upon these earlier insights, the present research capitalizes on the RNA-seq data from our prior work to further investigate the expression patterns of lncRNAs during mustard seed germination. This study not only advances our understanding of the complex transcriptional regulatory mechanisms involved in mustard's response to drought stress but also lays the groundwork for future endeavors aimed at enhancing drought tolerance in this economically significant crop.

## MATERIALS AND METHODS

### Plant materials and germination experiment

In a prior investigation, we have identified a leaf mustard cultivar ('WeiLiang', WL) demonstrating drought tolerance, along with a drought-sensitive one ('ShuiDong', SD), specifically during the germination phase (*Wei et al., 2023*). For this study, fresh seeds with high viability from WL and SD were selected as the plant materials. Seeds underwent an initial sterilization process with 75% ethanol for 3 min, followed by a triple wash with ddH$_2$O, and subsequent drying using absorbent paper. These seeds were then placed on double-layer filter paper within 10 × 10 cm plastic germination boxes (Wantong Medical Device company, Taixing, China), which were moistened with seven mL of ddH$_2$O (control, 0 Mpa) or polyethylene glycol 8000 (PEG8000) solution adjusted to achieve different osmotic potentials mimicking moderate (−0.5 MPa) or severe (−1.0 MPa) drought stress. PEG8000, chosen for its ability to influence osmotic pressure while being too large to be absorbed by plants, was first prepared according to Michel's formula (*Michel, 1983*) and then adjusted based on the true water potential determined at 25 °C using a Dew Point Microvolt meter HR-33T (Wescor, Logan, Utah, USA). Germination experiments were conducted for 7 d in plastic boxes within an artificial climate chamber (Model DLRX-350D-LED, Jinmin Instrument Equipment Company, Shanghai, China) maintained at 25 °C under a 12-h light/12-h dark photoperiod with a light intensity of 120 $\mu$mol m$^{-2}$ s$^{-1}$. Three biological replicates were performed for each treatment per cultivar. All plastic germination boxes were randomly positioned within the climate chamber, and to minimize micro-environmental effects, the positions of the boxes were rotated among shelves and locations within shelves daily following seed germination assessments.

## Hydrotime modeling and statistical analysis

The daily collected germination data from two mustard cultivars under different drought conditions in the aforementioned experiment were subsequently analyzed using a hydrotime model (*Bradford, 2005*). This model posits that the fixed constant, $\theta_H$, is determined by the product of the difference between the actual water potential ($\Psi$) and the base water potential [$\Psi_{b(g)}$], and the requisite germination time ($t_g$), mathematically expressed as:

$$\theta_H = [\Psi - \Psi_{b(g)}]t_g. \tag{1}$$

This equation implies a direct proportionality between the germination rate of seeds and the water potential, indicating accelerated seed germination with increasing water potential. Given the seed germination curve's close resemblance to a normal distribution, it can be linearized using a probit transformation, facilitating more convenient parameter estimation:

$$\text{Probit}(g) = [\Psi - \theta_H/t_g - \Psi_{b(50)}]/\sigma. \tag{2}$$

In this equation, $\Psi_{b(g)}$ adheres to a normal distribution, where the mean value corresponds to $\Psi_{b(50)}$—the base water potential associated with a seed germination rate of 50%—and the standard deviation is represented by $\sigma$.

## RNA-Seq

A total of 42 samples were obtained from two leaf mustard cultivars (WL and SD) at various time points during germination: 0 h (start of the experiment), 12 h, 24 h, and 36 h, under both control (0 MPa) and drought stress (−1.0 MPa) conditions. Each time point and condition included three biological replicates for total RNA extraction, utilizing the RNeasy Plant Mini Kit (Qiagen, Hilden, Germany). The quality and quantity of the extracted total RNA were assessed using an Agilent 2100 Bioanalyzer (Agilent, Palo Alto, CA, USA). Next, M-MuLV Reverse Transcriptase and DNA Polymerase I were utilized to synthesize the first- and second-strand of cDNA, respectively, as per the methodology outlined by *Agarwal et al. (2015)*. The double-stranded cDNA underwent end-repair, followed by the enrichment of cDNA fragments within the 370 bp to 420 bp range utilizing the AMPure XP system (Beckman Coulter Life Sciences, Brea, CA, USA). Subsequently, PCR amplification was carried out, and the resulting PCR products were purified to assemble the final library for sequencing, following the standard methodology. The concentration of the constructed library was precisely quantified using both a Fluorometer (Qubit 3.0, Thermo Fisher, Waltham, MA, USA) and qRT-PCR, following the procedures detailed by *Griffith et al. (2015)*. Subsequently, the library underwent paired-end sequencing on the Illumina NovaSeq 6000 platform (Novogene, Tianjin, China). The RNA-seq data generated have been deposited in the China National GeneBank DataBase (CNGBdb) under the reference number CNP0004113.

## Gene expression and functional analysis

The raw sequencing data underwent quality filtering using Fastp to eliminate low-quality sequences, followed by alignment of the filtered data to the mustard reference

genome sequence (http://brassicadb.cn/#/) using Hisat2 (*Chen et al., 2018*; *Kim et al., 2019*). Read counts were obtained through featureCounts (*Liao, Smyth & Shi, 2014*). Differential expression analysis was conducted using DEseq2 (*Love, Huber & Anders, 2014*), with the criteria for selecting differentially expressed genes set at FDR < 0.05 and $|\log_2(\text{FoldChange})| > 2$ (*Liao, Smyth & Shi, 2014*). Expression levels were quantified by calculating Fragments Per Kilobase of exon model per million mapped fragments (FPKM) values. Pfam annotations were acquired using hmmsearch and the Pfam database (http://pfam.xfam.org/). Furthermore, protein sequences were compared against the NCBI non-redundant protein sequences (NR), NCBI non-redundant nucleotide sequences (NT), and SWISS-PROT protein sequence databases using blastx and blastp. Gene Ontology (GO, http://geneontology.org/) and Kyoto Encyclopedia of Genes and Genomes (KEGG, https://www.kegg.jp/) annotations were obtained by mapping the database IDs.

## Identification of drought-responsive gene families

In the present study, two drought-responsive gene families, *Aquaporin* and *LEA*, were identified from the mustard reference genome utilizing HMMER3 (https://hmmer.janelia.org/) (*Mistry et al., 2013*). Aquaporin was characterized by the presence of a specific structural domain known as Major Intrinsic Protein (MIP) (PF00230). The LEA family was further classified into eight subfamilies: LEA_1 (PF03760), LEA_2 (PF03168), LEA_3 (PF03242), LEA_4 (PF02987), LEA_5 (PF00477), LEA_6 (PF10714), Dehydrin (PF00257), and SMP (PF04927). The Pfam-A dataset was obtained from PFAM (https://pfam.janelia.org/) using the online tool hmmscan (https://www.ebi.ac.uk/Tools/hmmer/search/hmmscan). Initially, the gene family was scanned using the domain file as the primary template and genes with an $E$-value below 1e−10 were retained. The filtered genes then served as secondary templates for a subsequent round of scanning. Genes from this second phase were similarly filtered with an $E$-value below 1e−10. Ultimately, the putative genes within the gene family were identified.

## LncRNA identification and differential expression analysis

The StringTie (*Pertea et al., 2015*) software was utilized to assemble the mustard transcriptome. Transcripts that overlapped with mRNA, rRNA, tRNA, snoRNA, and snRNA were excluded from further analysis. Additionally, transcripts shorter than 200 bp, with fewer than two exons, and showing low expression levels (<3 reads) were also excluded based on the exclusion criteria proposed by previous reports (*Derrien et al., 2012*; *Novikova, Hennelly & Sanbonmatsu, 2012*; *Dou et al., 2021*; *Ren et al., 2022*; *Zhang et al., 2022*). After filtering transcripts based on sequence length and read count, their coding potential was evaluated using two tools: the coding potential assessment tool (CPAT) (*Wang et al., 2013*) and the coding potential calculator (CPC) (*Kong et al., 2007*). The online tool Pfamscan (https://www.ebi.ac.uk/Tools/pfa/pfamscan/) was subsequently employed to search for potential protein domains within the candidate lncRNA transcripts, utilizing the HMM library. Transcripts that did not exhibit protein-coding ability were categorized as lncRNAs. The final set of lncRNAs identified in this study were those consistently predicted by all three methods. The analysis of differentially expressed lncRNAs (DEL) followed a methodology akin to that used for mRNA analysis (*Wei et al., 2023*).

## Target gene prediction and function enrichment analysis

The expression correlation between all mRNAs and lncRNAs was established based on the Pearson correlation coefficient. The Benjamini–Hochberg method was utilized to adjust the significance of $P$-values. LncRNA-mRNA pairs possessing an absolute correlation coefficient value exceeding 0.9 and a Q value less than 0.05 were then selected. Following this, the ASSA algorithm was employed to assess the potential of direct base pairing, either co-transcriptional or post-transcriptional, between the identified lncRNAs and mRNAs for functional interactions (*Antonov et al., 2018*). The ASSA results were further filtered based on the threshold of *FDR* < 0.05 to obtain the final list of target genes. For the identification of DEL target genes, GO, KEGG and Pfam enrichment analyses were conducted using the clusterProfiler package, adopting a $P$-adjusted value of 0.05 as the significance enrichment threshold (*Yu et al., 2012*).

## Weighted gene co-expression network analysis

The weighted gene co-expression network analysis (WGCNA) package was employed for co-expression network analysis of all lncRNAs, with the soft-thresholding parameter set to 6 (*Zhang & Horvath, 2005*). Hierarchical clustering tree construction within the network was accomplished based on the gene dissimilarity matrix, aiding in the identification of differentially expressed modules. Dynamic tree cutting was implemented for module allocation, utilizing default parameters, including a minimum module size of 30 and a cut height of 0.25.

## Verification of lncRNA expression by qRT-PCR

Real-time quantitative polymerase chain reaction (qRT-PCR) was employed to assess the expression levels of lncRNA in leaf mustard plants under a 36-hour drought stress treatment ($-1.0$ Mpa) during germination. Total RNA from the plants was isolated using the RNA Easy Fast Plant Tissue Kit (DP452) from Tiangen Biotech Co., Ltd. (Beijing, China). The SnapGene (v3.2.1) software was utilized for designing qRT-PCR primers specific to lncRNAs. Eight lncRNAs were randomly selected for qPCR, which targets *Aquaporin* and *LEA* gene families. The ABI Quant Studio 6 Fluorescence Quantitative PCR System (Applied Biosystems by Life Technologies, Waltham, MA, USA) was employed for quantitative analysis. The $2^{-\Delta\Delta CT}$ method was utilized to calculate the relative RNA expression levels (*Livak & Schmittgen, 2001*). Each group of data was assessed through three technical replicates, and standard deviations were calculated. The internal reference gene for lncRNAs was *BjuActin7*.

## Statistical analysis

The germination parameters such as the time to reach 50% of maximum germination (t50), the time required for the germination percentage to increase from 25% to 75% (U7525), and the area under the germination curve within 72 h (AUC) were inferred from the germination curve of each treatment separately, each with three replications. Then, these parameters are depicted as mean values, each with its respective standard deviation (S.D., $n = 3$). The identification of statistically significant differences (at $P < 0.05$) among varying treatment conditions was enabled by the utilization of Duncan's multiple range test,
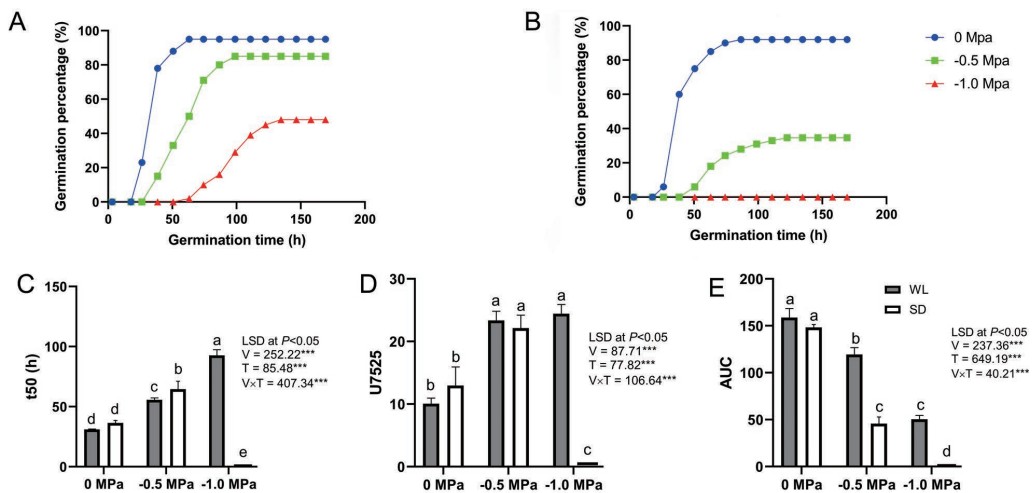

**Figure 1** **Seed germination dynamics of two leaf mustard cultivars ('WeiLiang', WL; and 'ShuiDong', SD) under various drought stress conditions.** (A) Germination curve of the drought-tolerant cultivar WL under various drought conditions (0 MPa, −0.5 MPa, and −1.0 MPa). (B) Germination curve of the drought-sensitive cultivar SD. (C) Comparison of t50 (the time to 50% maximum germination percentage) for the two cultivars under drought conditions estimated from the germination curve. (D) Comparison of U7525 (the time difference to reach 25% and 75% of the maximum germination percentage) for the two cultivars under drought conditions estimated from the germination curve. (E) Comparison of AUC (the area under the germination curve) for the two cultivars under drought conditions.

conducted *via* the SPSS software package, version 16.0 (SPSS Inc., Chicago, IL, USA). The creation of visual data representations was accomplished using GraphPad Prism version 9.0.0 and Microsoft Office Excel 2019.

# RESULTS

## Germination dynamics of leaf mustard cultivars

The seeds of the drought-tolerant leaf mustard cultivar (WL) and the drought-sensitive one (SD) were germinated under various water potential conditions in germination boxes. While there was no significant difference in the germination rate between the two cultivars under control conditions, WL showed a significantly higher germination rate than SD under drought stress (Figs. 1A and 1B). Additionally, WL demonstrated superior root length, shoot length, and fresh weight compared to SD under drought stress. This trend was also reflected in physiological indicators such as catalase (CAT) activity, peroxidase (POD) activity, superoxide dismutase (SOD) activity, and proline content (*Wei et al., 2023*).

As depicted in Fig. 1, germination parameters such as t50, U7525, and AUC were calculated based on the seed germination curve for each cultivar under different drought treatments. Statistical analysis revealed that seed germination was significantly affected by the cultivars (WL or SD), the drought treatments (0 Mpa, −0.5 Mpa, and −1.0 Mpa), and their interaction (Figs. 1C to 1E).

**Table 1 Hydrotime model parameters for two leaf mustard cultivars.** 'WL' represents the cultivar 'Weiliang' and 'SD' represents 'Shuidong'.

| Cultivar | $\Psi_{b(50)}$(MPa) | $\sigma_{\Psi b}$(MPa) | $\theta_H$(MPa h) |
|----------|------|------|------|
| WL | −1.073 | 0.339 | 43.054 |
| SD | −0.437 | 0.195 | 16.695 |

A lower t50 value signifies greater seed vigor. Under 0 MPa, there was no substantial difference between the two cultivars in t50 (31.06 h *vs* 36.41 h), indicating comparable germination rates for WL and SD. However, at −0.5 MPa, t50 significantly escalated to 55.72 h for WL and even higher (64.53 h) for SD, demonstrating notable discrepancies between the cultivars. Under −1.0 MPa, t50 rose further to 92.71 h for WL, suggesting that WL seeds germinated more sluggishly under severe drought stress, while SD did not germinate at all (Fig. 1C).

A smaller U7525 value indicates higher germination uniformity within the seed population. Under the control condition (0 Mpa), germination consistency between the two cultivars did not markedly differ, with U7525 ranging from 10.06 h to 12.96 h. However, under moderate drought stress (−0.5 Mpa), U7525 values rose significantly for both cultivars to 22.14 h and 23.36 h respectively, with no considerable difference between the two cultivars. Under severe drought stress (−1.0 MPa), the value for WL changed marginally (24.43 h), while SD failed to germinate (Fig. 1D).

The area under the curve (AUC) provides an overall assessment of seed germination quality, with a larger AUC indicating superior germination quality. There were no significant differences in AUC values between the two cultivars under control conditions. However, under moderate and severe drought stress, WL exhibited a significantly higher AUC value than SD, suggesting superior germination quality for WL and inferior quality for SD (Fig. 1E).

To explore deeper into the differences in drought tolerance among the cultivars, we developed a hydrotime model by fitting it to the germination data obtained under various drought treatments. Subsequently, we estimated key parameters to better understand the dynamics of drought performance in the cultivars. The results highlighted significant differences in the base water potential value $\Psi_{b(50)}$ of seed germination between the cultivars (Table 1). WL displayed a high stress-tolerance level, as exhibited by a very low $\Psi_{b(50)}$ value of −1.073 MPa. Conversely, SD was more drought-sensitive, since it showed a much higher $\Psi_{b(50)}$ (−0.437 MPa). Additionally, the required hydrotime constant $\theta_H$ for seed germination varied among the cultivars, ranging from 43.054 MPa h for WL to 16.695 MPa h for SD.

Collectively, our findings underscore distinct responses to drought stress between the cultivars, with WL exhibiting superior growth capability compared to SD. Hence, it is worthwhile to utilize RNA-seq data from both cultivars to further explore the influence of lncRNAs on their markedly different responses to drought stress.

## Identification and expression analysis of lncRNA

We previously used RNA-Seq to investigate the gene expression patterns linked to drought in leaf mustard during germination under drought conditions. The dataset comprised samples from two cultivars (WL, SD) gathered at four time points (0 h, 12 h, 24 h, and 36 h) during germination, under both control (0 Mpa) and severe drought stress (−1.0 Mpa) scenarios. The analysis led to the identification of numerous drought-inducible mRNAs (*Wei et al., 2023*). In this study, we further employed three distinct techniques (CPC, CPAT, and Pfam) to precisely identify and analyze lncRNAs. A total of 29,983 lncRNAs were successfully identified across all samples, as illustrated in Fig. 2. Contrasting with protein-encoding genes (mRNA), lncRNAs exhibited distinctive characteristics, notably fewer exons, with the majority having only two to four exons (Fig. 2A), and considerably shorter sequence lengths (Fig. 2B). Principal component analysis revealed distinct lncRNA expression patterns between the two mustard cultivars (Fig. 2E). The differential expression analysis highlighted time-specific expression dynamics of lncRNAs in leaf mustard. The number of DELs (differentially expressed lncRNAs) under the control condition (0 MPa) showed a gradual increase over time at 12 h, 24 h, and 36 h post-germination, in comparison to the initial time point (0 h) for SD. Moreover, the upregulated DELs at these time points numbered 872, 1,090, and 1,356, while the downregulated DELs were 796, 986, and 1,165, respectively (Fig. 2C). Conversely, under severe drought conditions (−1.0 Mpa) for SD, the upregulated DELs decreased to 517, 684, and 691, respectively, while downregulated DELs were limited to 478, 549, and 567, respectively. Out of these, 1,145 and 519 DELs exhibited consistent expression patterns at all three time points under both control and drought stress conditions, respectively (Figs. 3A and 3B).

The expression patterns of lncRNAs varied significantly between the two cultivars. In the cultivar WL, the quantity of DELs under control conditions at 12 h, 24 h, and 36 h post-germination compared to 0 h of germination was notably higher (Fig. 3C). Furthermore, under drought stress, WL showcased a greater number of DELs compared to SD at the same time points (Fig. 3D). These findings suggest that lncRNAs may play a more critical role in drought stress response in the drought-resilient cultivar WL.

The DELs under drought stress relative to the control condition exhibited variations across different time points and between cultivars. Specifically, only three DELs showed consistency at all three time points in SD (Fig. 3E), while in WL, merely two DELs demonstrated consistency at the same time points (Fig. 3G). A substantial number of DELs was observed between the two cultivars at three time points under drought stress, with a notable 1,256 lncRNAs manifesting differential expression at all three time points (Fig. 3F). Overall, the lncRNAs analysis across different time points and cultivars not only revealed their temporal specificity in expression but also highlighted marked pattern discrepancies between drought-tolerant and drought-sensitive cultivars.

## Prediction and functional annotation of lncRNA target genes

We delved into exploring the potential biological functions of the identified lncRNAs by investigating their target genes (mRNAs), to elucidate the distinct regulatory roles of lncRNAs in mustard cultivars WL and SD. Through Pearson correlation analysis of

Peer J

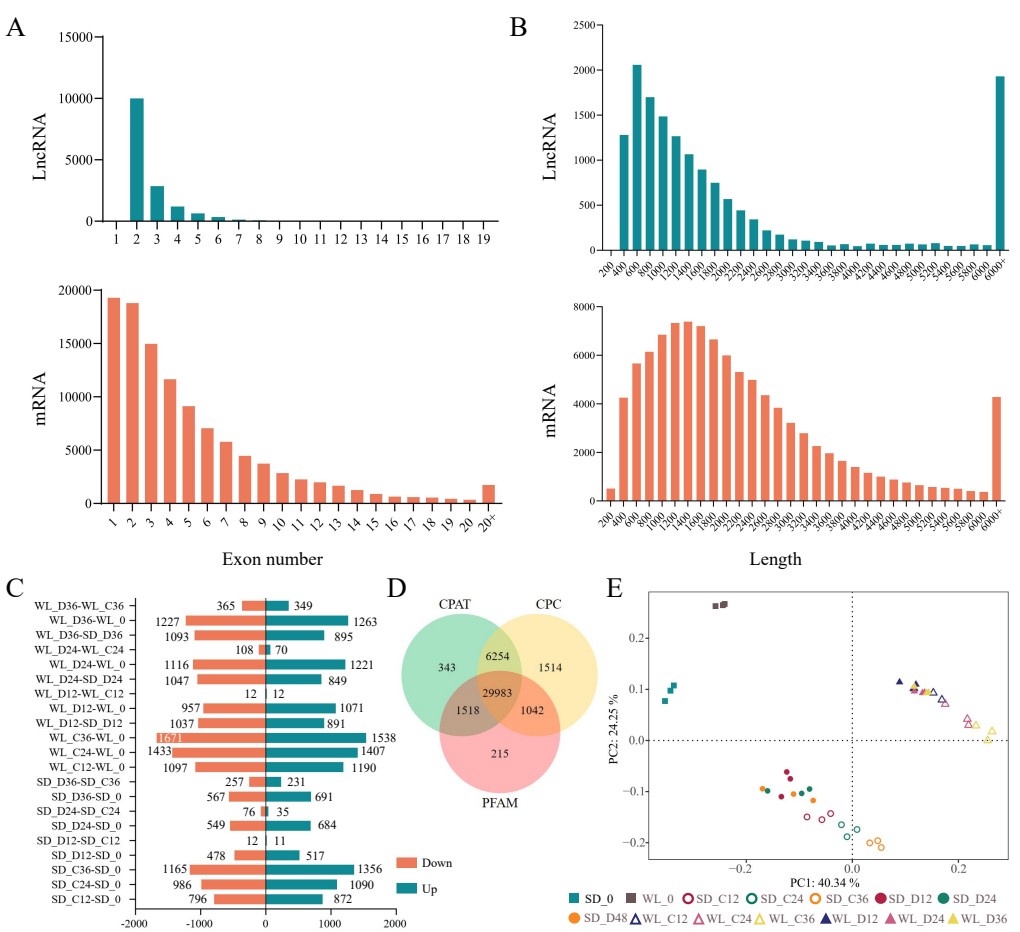

**Figure 2 Identification and expression analysis of lncRNAs.** (A) Histogram depicting the distribution of exonic counts for both lncRNAs and mRNAs; (B) Histogram illustrating the length distribution of transcripts for lncRNAs and mRNAs; (C) Number of differentially expressed lncRNAs under various treatment conditions; (D) Venn diagram depicting the overlap of lncRNAs identified by three distinct methods: Coding Potential Assessment Tool (CPAT), Coding Potential Calculator (CPC), and Pfam; (E) principal component analysis of lncRNA expression levels across 42 samples. WL refers to the cultivar 'Weiliang', while SD refers to the cultivar 'Shuidong'. C12, C24, and C36 denote germination at 12 h, 24 h, and 36 h under control conditions, respectively. Similarly, D12, D24, and D36 represent germination under drought conditions at the corresponding time points.

lncRNA and mRNA expression levels, coupled with RNA-RNA interaction predictions, we identified a total of 2,087 DELs and 12,433 corresponding target genes. Subsequent GO enrichment analysis of these target genes revealed distinct patterns across various comparison groups. Notably, in WL, the target genes at 12 h post-germination under drought stress exhibited enrichment in various GO terms such as "response to high light intensity" (GO:0009644), "plant-type vacuole membrane" (GO:0009705), and "seed oil body biogenesis" (GO:0010344) (Fig. 4A). Conversely, in SD, the target genes did not show enrichment in any GO terms during the same time frame. This disparity indicates a quicker response of genes associated with these functions in WL compared to SD during the initial stages of germination under drought stress. Furthermore, at 36 h of germination,

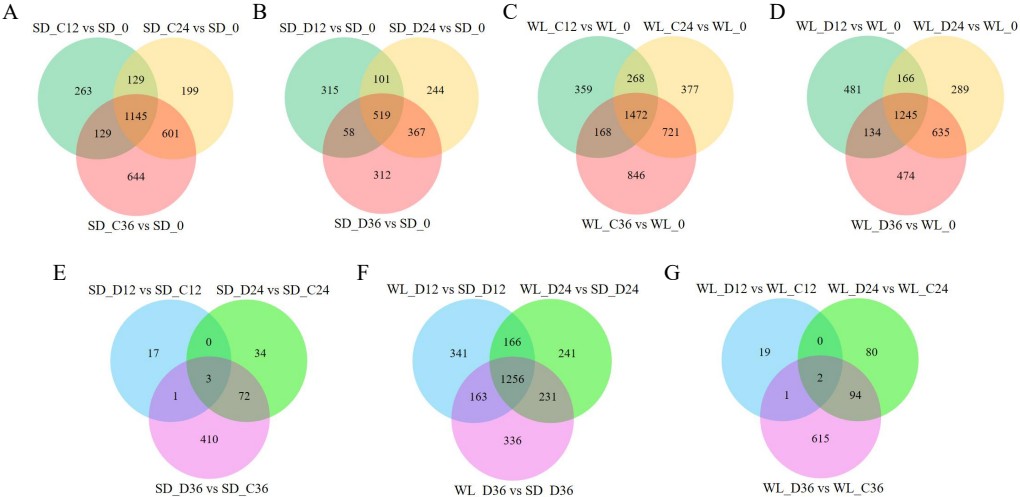

**Figure 3 Differential expression gene venn diagrams.** (A) Differential expression analysis of lncRNAs in the 'Shuidong' (SD) cultivar under control (C) conditions at 0 h, 12 h, 24 h, and 36 h of germination; (B) Differential expression analysis of lncRNAs in SD under drought (D) conditions at 0 h, 12 h, 24 h, and 36 h of germination; (C) Differential expression analysis of lncRNAs in the 'Weiliang' (WL) cultivar under control (C) conditions at 0 h, 12 h, 24 h, and 36 h of germination; (D) Differential expression analysis of lncRNAs in WL under drought (D) conditions at 0 h, 12 h, 24 h, and 36 h of germination; (E) Differential expression analysis of lncRNAs in SD under control (C) and drought (D) conditions at 12 h, 24 h, and 36 h of germination; (F) Differential expression analysis of lncRNAs between WL and SD under drought (D) conditions at 12 h, 24 h, and 36 h of germination; (G) Differential expression analysis of lncRNAs in WL under control (C) and drought (D) conditions at 12 h, 24 h, and 36 h of germination.

the enrichment level of "water transport" (GO:0006833) in WL was slightly higher than in SD, suggesting potential differences in the water transport mechanisms between the two cultivars under drought stress conditions.

Pfam enrichment analysis of target genes produced similar findings. The enrichment analysis outcomes revealed a noteworthy enrichment of MIP in both cultivars under drought stress relative to the control condition at 36 h of germination. Additionally, several LEA family members (Dehydrin, LEA_4, and SMP) demonstrated variable degrees of enrichment (Fig. 4B). Further, the KEGG pathway analysis highlighted the enrichment of several pathways related to drought stress in WL. Significantly, pathways such as photosynthesis (map00195) demonstrated remarkable enrichment at 36 h of germination in WL compared to SD under drought stress, reflecting similar findings observed for other metabolic pathways like nitrogen metabolism (map00910) and porphyrin and chlorophyll metabolism (map00860) (Fig. 4C).

## Co-expression analysis of lncRNAs and their target genes

The aforementioned analysis highlighted the potential for multiple lncRNAs to regulate the same target gene (mRNA), suggesting functional similarities among lncRNAs with similar expression patterns. To identify the key target genes of lncRNAs, we employed WGCNA to analyze the top 25% of lncRNAs based on the FPKM values variance. Utilizing a soft threshold of 6, we identified 3,651 lncRNAs, which were classified into 10 modules.

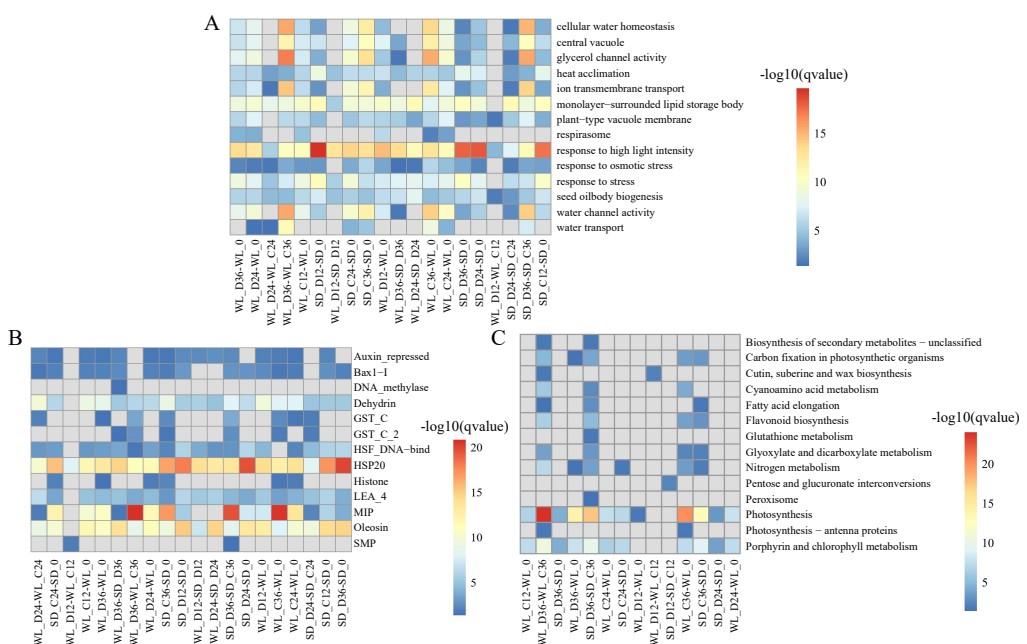

**Figure 4  Enrichment analysis results of differentially expressed lncRNA target genes under various drought stress conditions.** (A) Gene ontology (GO) categories; (B) Pfam protein families; (C) KEGG pathways.

Among these, the turquoise module exhibited the highest number of lncRNAs, totaling 1,569 (Fig. 5A). These lncRNAs collectively regulated 1,797 target genes (Fig. 5B). Notably, the green module, consisting of only 217 lncRNAs, displayed regulation of 3,929 target genes, suggesting instances of a single lncRNA modulating multiple target genes within this module. Similar observations were made for the blue module (655 lncRNAs regulating 4,129 target genes) and the brown module (562 lncRNAs regulating 17,486 target genes).

Correlation analysis between the expression levels of lncRNAs within each module and mRNAs from seed samples revealed module-specific gene expression patterns. For instance, both the brown and yellow modules exhibited strong correlations with transcript levels in seeds at the initial germination stage (0 h) in both cultivars. The turquoise module showed a significant correlation only with mRNA levels in seeds of WL at the initial germination stage. Furthermore, the red module showed a significant correlation with mRNA levels in both cultivars under drought stress at 12 h of germination, with a particularly strong correlation in WL compared to SD (Fig. 5F). These results highlight the distinct relationships between lncRNA expression within modules and mRNA abundance in different seed samples and under varying stress conditions.

To understand the functions of mRNA targeted by lncRNAs within each module, we conducted GO enrichment analysis for the target genes of lncRNAs across all modules. Except for the black and turquoise modules, which did not show significant enrichment of GO terms, the other modules demonstrated varying degrees of enriched GO terms (Table S1). Visualization through bubble charts depicting the enrichment analysis of

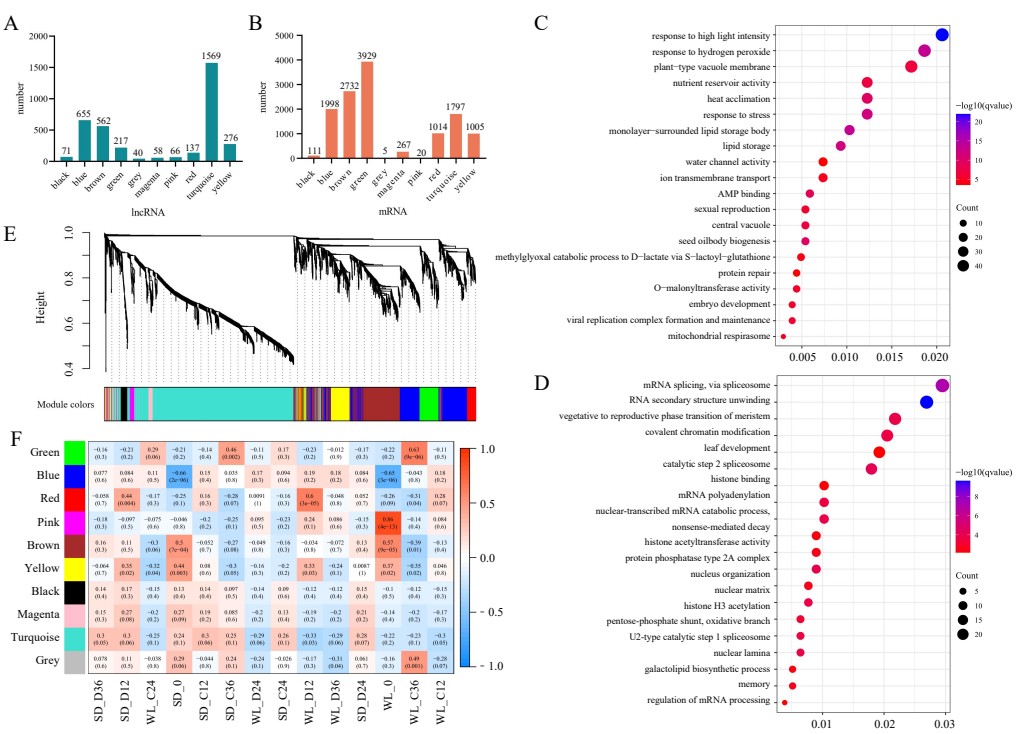

**Figure 5** **WGCNA Analysis of lncRNA expression.** (A) The number of lncRNAs in each lncRNA expression module; (B) The number of target genes for lncRNAs within these modules; (C) Gene Ontology (GO) enrichment results for target genes of lncRNAs in the brown module (top 20 sorted by Q-value); (D) GO enrichment results for target genes of lncRNAs in the red module (top 20 sorted by Q-value); (E) Hierarchical clustering tree of lncRNAs based on dissimilarity in adjacency relationships (dendrogram); (F) Correlation analysis between lncRNA expression modules and sample groups.

the brown module (Fig. 5C) and the red module (Fig. 5D) unveiled distinct patterns of enrichment. In the brown module, the enrichment analysis unveiled multiple stress response-related GO terms, including responses to high light intensity (GO:0009644), heat acclimation (GO:0010286), and stress responses (GO:0006950). Moreover, GO terms related to seed development such as seed oil body biogenesis (GO:0048316), nutrient reservoir activity (GO:0045735), and embryo development (GO:0009790) were also identified (Fig. 5C). On the contrary, the red module exhibited enrichment in signal processing such as mRNA splicing *via* spliceosome (GO:0000398), RNA secondary structure unwinding (GO:0010501), and vegetative to the reproductive phase transition of the meristem (GO:0010228) (Fig. 5D). These findings shed light on the specific functional roles of lncRNAs within different modules and provide insights into some key biological processes influenced by their interactions with target mRNAs.

## Identification and construction of regulatory network for drought-responsive genes

Pfam enrichment analysis unveiled the diverse regulatory roles of MIP (Aquaporin) and LEA proteins (dehydrin, LEA_4, and SMP) in leaf mustard's response to drought stress.
Aquaporins and LEA proteins have crucial functions in regulating seed germination and stress responses. Hence, we conducted a comprehensive analysis of these gene families, identifying *Aquaporins* and *LEA* (including 8 subfamilies) as pivotal gene families associated with seed germination and drought stress in mustard.

Within the *Aquaporin* gene family, we identified a total of 76 aquaporin genes, among which 39 were regulated by 83 lncRNAs, showing differential expression in at least one comparative group (Fig. 6A, Table S2). Among the lncRNAs regulating aquaporins, MSTRG.173897 exhibited consistently high expression levels in both cultivars after germination. Conversely, MSTRG.168716, MSTRG.163239, and MSTRG.1301 displayed lower expression levels at 12 h, 24 h, and 36 h of germination in both cultivars. Meanwhile, MSTRG.139136 demonstrated gradually increasing expression over time but exhibited relatively low expression under drought stress compared to the control condition in both cultivars. Notably, certain lncRNAs exhibited distinct expression patterns between the two cultivars under drought stress. For instance, the expression levels of MSTRG.164194, MSTRG.74300, and MSTRG.82428 in SD were higher than those in WL, potentially contributing to the observed differences in drought tolerance between the two cultivars.

We also identified 17 genes belonging to the *SMP* subfamily of the *LEA* family, regulated by 41 lncRNAs (Fig. 6B). The expression levels of these lncRNAs decreased during the seed germination stage but were higher under drought stress compared to the control condition. This suggests that these lncRNAs elevate expression levels under drought stress to regulate *SMP*. MSTRG.22298 exhibited the highest expression level and continued to express during the germination stage, with no significant difference observed between the control conditions of the two cultivars. However, the expression level of this lncRNA under drought stress in WL was significantly lower than in SD, indicating its potential role as a crucial negative regulator of drought tolerance in mustard.

For other *LEA* subfamilies, we identified 106 lncRNAs that regulate 17 *Dehydrin* genes, 66 lncRNAs controlling 9 *LEA _1* genes, 121 lncRNAs manipulating 141 *LEA _2* genes, six lncRNAs guiding 13 *LEA _3* genes, 70 lncRNAs governing 23 *LEA _4* genes, 52 lncRNAs managing eight *LEA _5* genes, and two lncRNAs overseeing seven *LEA _6* genes (Fig. S1 and Table S3). Analogous to *SMP*, these lncRNAs exhibited high expression during seed germination. Certain lncRNAs displayed consistent expression across all samples, with a noticeable downregulation in their expression levels in WL relative to SD under drought stress conditions, suggesting a negative regulatory role.

The network diagram depicting lncRNAs regulating target genes reveals a complex regulatory network, with multiple lncRNAs simultaneously regulating one or more *aquaporin* genes and *LEA* genes. For example, MSTRG.107159 simultaneously regulates six *PIP* genes, one of which is regulated not only by MSTRG.107159 but also by MSTRG.121325, MSTRG.130644, and MSTRG.26412. Similar regulatory relationships were observed in the *SMP* subfamily, with multiple lncRNAs regulating a single gene. Specifically, 18 lncRNAs simultaneously regulate *SMP-4*, and among these, 5 lncRNAs also regulate *SMP-3*. Comparable regulatory relationships were observed in several other subfamilies (Fig. S1).

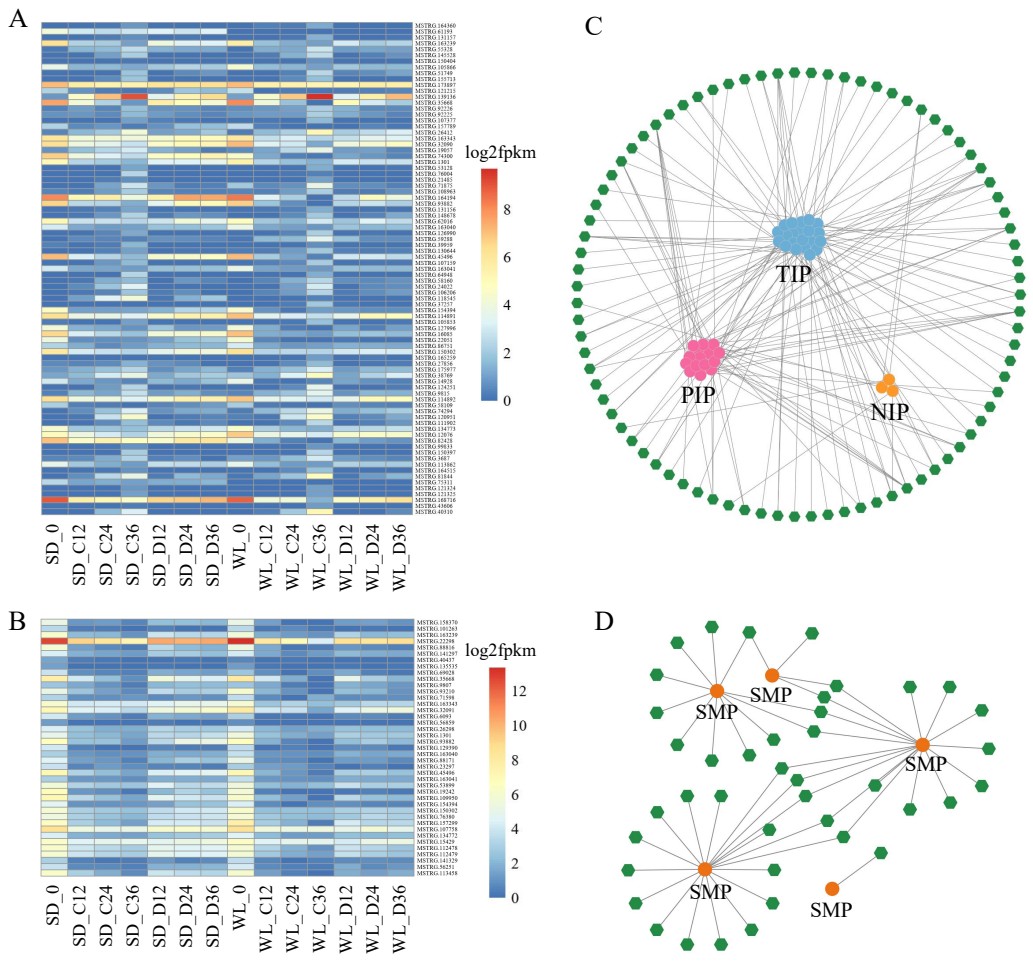

**Figure 6** **LncRNA regulatory network illustrating the interactions between Major Intrinsic Protein (MIP) and Seed Maturation Protein (SMP) genes.** A heat map depicting the expression profiles of lncRNAs regulating MIP (A) and SMP (B) families in 14 experimental groups. Network diagrams illustrating the lncRNA-mediated regulation of MIP (C) and SMP (D) families, where green hexagons represent distinct lncRNAs and circular dots of different colors represent genes.

## lncRNA regulates photosynthesis pathways

The KEGG pathway analysis has emphasized the critical role of the photosynthesis pathway (map00195) in the drought response in WL (Fig. 4C). Subsequently, we conducted a detailed analysis revealing the regulatory influence of lncRNAs on multiple target genes within this pathway (Fig. 7 and Fig. S2). During drought stress, the expression levels of lncRNAs were notably downregulated at 36 h post-germination in both mustard cultivars when compared to the control condition. Remarkably, a higher proportion of lncRNAs experienced downregulation in the drought-tolerant cultivar WL than in the drought-sensitive SD. This observation suggests that WL responds to drought stress by modulating genes related to photosynthesis through the suppression of a larger number of lncRNAs.

Several lncRNAs were identified as key regulators of genes involved in photosynthesis (Fig. 7). Notably, Psb27 was targeted by 20 lncRNAs, with 18 of them showing significant

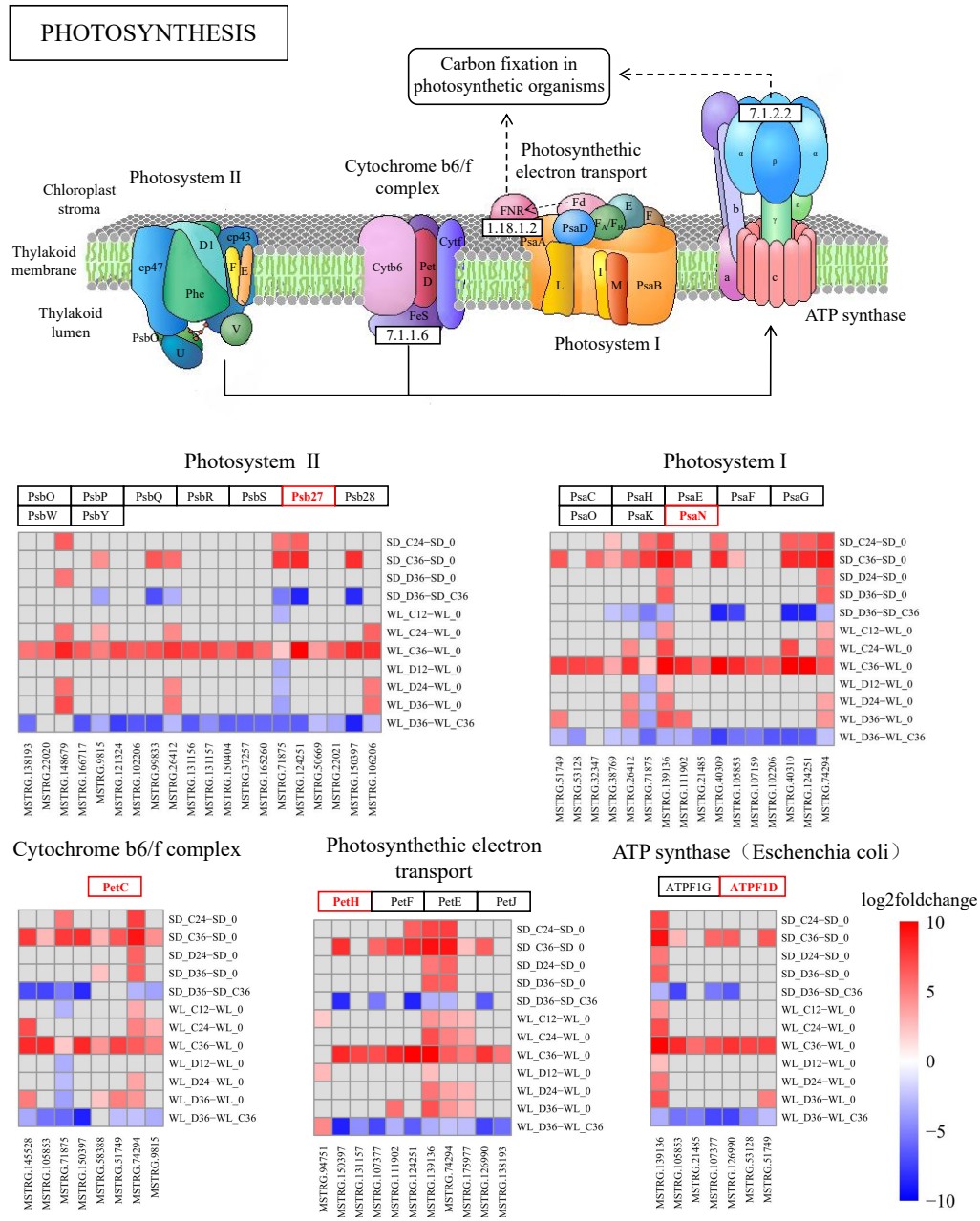

**Figure 7** KEGG enrichment analysis and lncRNA regulation within the photosynthetic pathway. lncRNA-mediated modulation of genes associated with photosynthesis and enrichment of log2FoldChange values in the photosynthetic pathway.

down-regulation in WL following 36 h of drought treatment, while only six exhibited a similar trend in SD under the same conditions. This regulatory trend by lncRNAs was also observed in other genes within the photosynthesis pathway. Among the lncRNAs influencing Psb27, MSTRG.150397 stood out for its substantial down-regulation (8.78-fold). Furthermore, MSTRG.150397 was found to modulate *PetC*, *PetH*, and *PsbW*, all

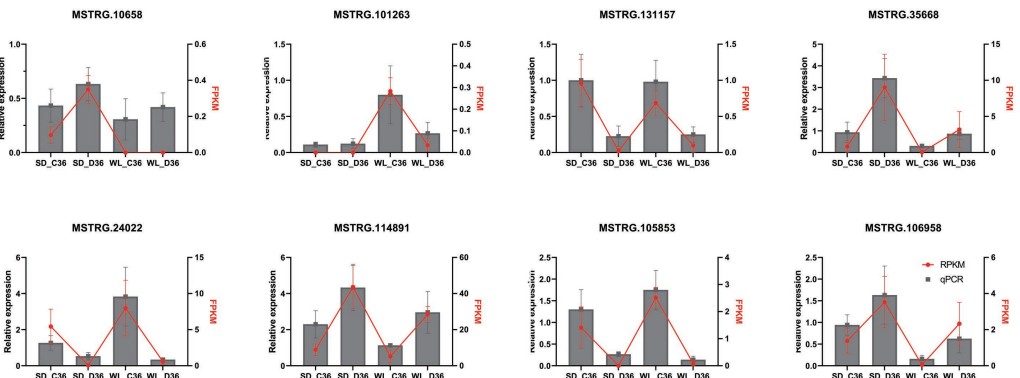

**Figure 8** **QRT-PCR verification of lncRNA expression.** SD refers to the drought-susceptible cultivar 'Shuidong,' while WL represents the drought-tolerant cultivar 'Weiliang'. C36 denotes the time point of 36 h post germination under control conditions, whereas D36 indicates the same time point under drought conditions. The relative gene expression levels were normalized to the expression of the internal reference gene *BjuActin7* and compared to the values quantified by RNA-Seq, expressed as fragments per kilobase of transcript per million mapped reads (FPKM).

of which displayed significant down-regulation in WL after 36 h of drought treatment. Additionally, the expression levels of lncRNA in the remaining samples showed varying degrees of down-regulation compared to the initial germination stage.

## Verification of lncRNA by qRT-PCR

To validate the identified differential expression of lncRNAs from the genome-wide RNA-seq analysis, qRT-PCR was performed on eight randomly selected lncRNAs linked to the *Aquaporin* and *LEA* gene families (Fig. 8), utilizing specific primers detailed in Table S4. The results indicated that four lncRNAs (MSTRG.101263, MSTRG.131157, MSTRG.24022, and MSTRG.105853) displayed down-regulation under drought stress conditions across both cultivars, while the remaining four (MSTRG.10658, MSTRG.35668, MSTRG.114891, MSTRG.106958) exhibited up-regulation. The expression patterns observed *via* qRT-PCR were consistent with the RNA-seq data, affirming the reliability and robustness of the lncRNA identification process.

## DISCUSSION

Numerous studies have underscored the crucial role lncRNA plays in plant development, growth, and stress resistance by managing their target mRNAs (*Chen et al., 2021*). The functionality of lncRNAs is not isolated but rather, they indirectly orchestrate physiological processes in plants through interactions with additional molecules (*Guan et al., 2024*). In the present investigation, advanced high-throughput sequencing methodologies were utilized to scrutinize the transcriptome of two distinct leaf mustard cultivars, exhibiting contrasting drought tolerance phenotypes at the germination phase, subsequently constructing a regulatory network comprising lncRNA and mRNA. A total of 29,983 lncRNAs was identified from a pool of 42 leaf mustard samples, a count significantly surpassing that detected in upland rice (*Yang et al., 2022*), a discrepancy which may be

attributable to the larger assortment of cultivars and treatments incorporated in this study. Mirroring previous studies, most identified lncRNAs were categorized as long intergenic lncRNA (lincRNA) (*Zhang et al., 2014*). The core attributes of lncRNA and mRNA were further examined, revealing that lncRNA sequences exhibited markedly shorter lengths compared to mRNA counterparts (Fig. 2). Furthermore, lncRNA displayed a paucity of exons and lower expression levels, findings that corroborate previous findings (*Chen et al., 2021*; *Xu et al., 2021*). To elucidate the functional capacity of lncRNAs, target genes for the 2,087 DELs were computationally predicted, yielding a total of 12,433 target genes, which were found to be prominently associated with photosynthesis and hormone metabolism pathways. Therefore, this study suggests that drought stress profoundly influences plant hormone metabolism and photosynthesis, a conclusion in alignment with prior studies (*Chaves et al., 2002*).

As a fundamental physiological process in plant growth and development, photosynthesis plays a pivotal role in the response to abiotic stress (*Longo et al., 2020*). In maize, seeds can modulate photosynthesis during germination to withstand abiotic stress (*Meng, Wen & Zhang, 2022*). Under abiotic stress conditions during seed germination, differentially expressed genes (DEGs) in *Apocynum venetum* and differentially expressed proteins in mulberry (*Morus alba* L.) are primarily concentrated in photosynthesis-related pathways (*Li et al., 2023*; *Wang et al., 2023*). Also, our study reveals significant enrichment of the photosynthesis pathway at 36 h post germination (Fig. 7), paralleling discoveries made during the germination of quinoa (*Chenopodium quinoa*) seeds (*Hao et al., 2022*). This suggests a critical role of photosynthesis in the mustard plant's response to drought stress during seed germination. Within this pathway, we detected numerous target genes, regulated by lncRNAs, that are associated with drought stress. *Psb27* contributes to the efficient assembly and repair of PSII under stress conditions (*Huang et al., 2021*). *PsaN* and *PetC* are central genes in the response to drought (*Arab et al., 2022*). The expression levels of genes related to photosynthetic electron transport (*PetE* and *PetF*) significantly decrease under stress conditions (*Wang et al., 2021*). In our study, these stress response genes, regulated by lncRNAs in the photosynthesis pathway, exhibit significant upregulation or downregulation across the two cultivars (Fig. 7). This implies that lncRNAs and their target genes in the photosynthesis pathway are connected to drought resistance in mustard and could represent key genes that account for the differences in drought resistance among different mustard cultivars.

Water supply is a determining factor for seed germination and seedling emergence, with plant aquaporin proteins playing an instrumental role in controlling water transmembrane transport and seed germination regulation. Despite their importance, studies on mustard aquaporin protein genes are scarce. In the present study, the Pfam enrichment analysis of lncRNA target genes implies that genes associated with aquaporin protein and *LEA* gene families may serve a vital role in regulating mustard seed germination under drought stress. Aquaporin proteins in plants are typically categorized into four subfamilies, *i.e.,* PIP, TIP, NIP, and SIP (*Maurel et al., 2015*). We identified 39 aquaporin protein genes regulated by 83 lncRNAs (Fig. 6). Certain lncRNAs demonstrate differential expression across the two cultivars and are implicated in the regulation of genes associated with drought stress.

 

For instance, the gene *TIP2;1* (two copies in the mustard genome, *i.e., Bjuva01g39180* and *Bjuvb01g38360*), previously recognized as a key gene in drought stress response (*Lukšić et al., 2023*), is regulated by a lncRNA, MSTRG.139136. *TIP3;1* (three copies in the genome, *i.e., Bjuvb05g55980*, *Bjuva07g3020*, and *Bjuvb03g35190*) is recognized as a positive regulator of the abscisic acid (ABA) response. Mutants of *TIP3;1* have been shown to diminish the critical water potential necessary for seed germination (*Footitt et al., 2019*). Here we showed that this gene is simultaneously regulated by three lncRNAs (MSTRG.45496, MSTRG.164194, and MSTRG.168716). It is worth mentioning a significant variation in the expression level of *TIP3;1* between the two cultivars, proposing its potential role as a pivot gene during drought stress response in mustard seed germination. *LEA* genes also contribute to plant stress responses associated with water scarcity (*Graether, 2022*). *RAB28* (BjuVB01G31660), a member of the LEA_5 subfamily, enhances seed germination (*Guan et al., 2024*) and strengthens plant resilience to drought stress (*Borrell et al., 2002*; *Amara et al., 2013*) and is regulated by MSTRG.22298, which showed significant upregulation under drought stress conditions. Taken together, the recognition of these genes augments our comprehension of the mustard drought-related gene families and regulatory networks, offering valuable insights into the causes of the differences in drought resistance among various mustard varieties.

## CONCLUSIONS

We have successfully elucidated the pivotal role of lncRNAs in the response of leaf mustard to drought stress during seed germination, by employing high throughput RNA sequencing techniques. Through comprehensive analysis, a repertoire of drought-related lncRNAs and mRNAs were identified. Subsequent GO, Pfam enrichment, KEGG pathway analyses, and network construction of lncRNA and their target genes revealed their involvement in the regulatory processes underlying mustard's tolerance to drought stress. Notably, MSTRG.150397 and MSTRG.150397 were found to potentially modulate target genes (*Psb27*, *PetC*, *PetH*, and *PsbW*) implicated in the photosynthesis pathway, thereby impacting the plant's ability to withstand drought stress. Additionally, MSTRG.107159 was identified as a regulator of six drought-responsive *PIP* genes. This research provides a solid foundation for further investigations concerning the potential roles of lncRNAs in enhancing plant drought resistance. Furthermore, it offers novel genetic resources that could potentially be harnessed for the genetic engineering of crop species with improved drought tolerance.

### Funding

This research was funded by the Natural Science Foundation of Guangdong Province, China (No. 2022A1515011837), the National Natural Science Foundation of China (No. 32072131), the Guangdong-Hong Kong-Macao Greater Bay Area Teaching Education Project (No. WGKM2023116), the Guangdong University of Petrochemical Technology

School-Level Educational Reform Project (No. JY202224), and the Projects of Talents Recruitment of GDUPT (2021rc009). The funders had no role in study design, data collection and analysis, decision to publish, or preparation of the manuscript.

## Grant Disclosures

The following grant information was disclosed by the authors:
Natural Science Foundation of Guangdong Province, China: No. 2022A1515011837.
National Natural Science Foundation of China: No. 32072131.
Guangdong-Hong Kong-Macao Greater Bay Area Teaching Education Project: No. WGKM2023116.
Guangdong University of Petrochemical Technology School-Level Educational Reform Project: No. JY202224.
Projects of Talents Recruitment of GDUPT: 2021rc009.

## Competing Interests

The authors declare there are no competing interests.

## Author Contributions

- Jinxing Wei performed the experiments, prepared figures and/or tables, and approved the final draft.
- Haibo Li performed the experiments, prepared figures and/or tables, and approved the final draft.
- Xiaoer Huang performed the experiments, prepared figures and/or tables, and approved the final draft.
- Yongguo Zhao analyzed the data, prepared figures and/or tables, and approved the final draft.
- Lejun Ouyang analyzed the data, prepared figures and/or tables, and approved the final draft.
- Mingken Wei analyzed the data, prepared figures and/or tables, authored or reviewed drafts of the article, and approved the final draft.
- Chun Wang analyzed the data, authored or reviewed drafts of the article, and approved the final draft.
- Junxia Wang conceived and designed the experiments, authored or reviewed drafts of the article, and approved the final draft.
- Guangyuan Lu conceived and designed the experiments, authored or reviewed drafts of the article, and approved the final draft.

## DNA Deposition

The following information was supplied regarding the deposition of DNA sequences:
The sequences are available at the China National GeneBank DataBase: CNP0004113, DOI 10.26036/CNP0004113.

## Data Availability

The raw measurements are available in the Supplementary Files.

## Supplemental Information

Supplemental information for this article can be found online at http://dx.doi.org/10.7717/peerj.17661#supplemental-information.

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
