# Peer review of "Elucidating the regulatory role of long non-coding RNAs in drought stress response during seed germination in leaf mustard"

_PeerJ, doi:10.7717/peerj.17661_

## Round 0.1 · original submission · Major Revisions

Drought stress represents a significant environmental challenge, necessitating a comprehensive understanding of crop responses and strategies to mitigate its adverse effects. Hence, each investigation aimed at enhancing our comprehension of drought stress is of paramount importance. I am confident that the insights gleaned from your research will prove invaluable to future scholars. Nonetheless, meticulous attention to technical details is imperative for refining your article. I strongly advocate for a comprehensive review of the reviewers' feedback and a thoughtful evaluation of each recommendation. In instances where disagreement arises, providing cogent justifications for dissenting viewpoints would enhance the clarity and credibility of your arguments.

**Language Note:** The review process has identified that the English language must be improved. PeerJ can provide language editing services - please contact us at [email protected] for pricing (be sure to provide your manuscript number and title). Alternatively, you should make your own arrangements to improve the language quality and provide details in your response letter. – PeerJ Staff

·

Basic reporting

I have suggested a few minor revisions in the article.

Experimental design

Experiments were prepared appropriately.The findings are written clearly.

Validity of the findings

The findings are written clearly.

·

Basic reporting

no comment

Experimental design

no comment

Validity of the findings

no comment

Additional comments

In the manuscript (Elucidating the regulatory role of long non-coding RNAs in drought stress response during germination in leaf mustard), the authors represent the first investigation into the response of lncRNAs in leaf mustard to drought stress during seed germination. IncRNAs of two mustard varieties WL and SD were studied. KEGG enrichment analysis and Pfam enrichment analysis of mustard provide valuable reference for cultivating drought resistant mustard varieties. However, I think the manuscript should be revised before publication.

Major:
1. The main shortcoming of the current manuscript is that the language level is not very fluent, and some minor errors exist. I suggest that the authors could turn to a language editing service company or a fluent speaker.
2. The authors have made great effort in identifying lncRNAs and their potential mRNAs, then what’s the conclusion of the manuscript? Which gene or lncRNA contribute to the differential response/tolerance of the two leaf mustard varieties? The authors should give more deeply excavation or explanation.
3. I have noticed that the author has already conducted three replicates in the experiment for RNA-seq and LncRNAs. However, I still suggest that the authors could carry out qRT-PCR analysis to further support the accuracy of the results.
4. The discussion part seems like a repetition of the results. The part should be improved to take the meaning of the findings further.

Minor:
1. The abstract part did not consist of important findings from the large work.
2. The tense of many narratives throughout the article is inconsistent. For example, lines 108-110
3. Lines 35-38, what is the main meaning? The authors should rewrite it.
4. Lines 54-55, is particularly susceptible to drought and results in reduced yield.
5. Line 217: dynemics or dynamics?
6. Line 259: 0underscore?
7. The figure legends between figures 5 & 6 did not seem completely corresponding.

Reviewer 3 ·

Basic reporting

MS No. #94760
Topic: Elucidating the regulatory role of long non-coding RNAs in drought stress response during germination in leaf mustard
Dear Editor,
I have reviewed the MS with above-mentioned details.
The authors have taken up a good theme for the study but failed to express it in the form of a manuscript. The quality of English and grammar is good and suffers no major issues.
However, there are many issues associated with the quality of manuscript and plant-based understanding of the authors.
The major ones are as:
1. Authors have no clarity about the stage specific metabolism of mustard seeds. They seem confuse with last growth stage of seeds at the time of harvesting and prior to the germination.
2. Authors failed to express that what were the germination/growth conditions for seeds? Germination bed, material, moisture level, water potential of control solution, soil culture or hydroponics, etc. What was the temperature, humidity, light or dark period (if considered),
3. Importantly, authors failed to mention how drought was created? This parameter is crucial because objective of entire study is based on this theme.
4. No proper mention of methodology how RNAs were extracted and treated for generating data.
5. I have made many corrections directly in the attached MS.
Figure 6: I totally failed to understand why there is mention of Thermosynechococcus elongatus and Eschenchia coli (no such in my knowledge so far) in the Figure 6? Explain enrich the readers about it if I am ignorant about the same. If there is any typographical mistake, what these names have to do with the Mustard? If this picture is taken from the internet source, this seems not appropriate.
Authors have to address the issues raised.
Recommendation: Major revision

Experimental design

See above

Validity of the findings

See above and comments in the MS

Annotated reviews are not available for download in order to protect the identity of reviewers who chose to remain anonymous.

·

Basic reporting

The quality of English needs to be improved and authors are advised to check their manuscript with a fluent English speaker. e.g. Ensure consistency of terminology throughout the introduction. For example, "drought-tolerant" and "drought-sensitive" are hyphenated in some cases and not in others. Consistent hyphenation and terminology improves readability.

Experimental design

Methods not sufficiently described I could not find any information on the RNA extraction method or the sequencing profile or criteria for the identification of LncRNA..

Validity of the findings

This study provides excellent information about this neglected plant. The results and findings are presented in a nice way. Although the discussion is really disorganised and does not agree with the author's findings,the discussion should mainly focus on the research findings and consolidate their findings with other similar research rather than re-explaining the findings.
the authors must draw conclusions, which should be based on the objectives, method and material, results, discussion and future perspectives.

Additional comments

1) Although the abstract mentions the number of differentially expressed lncRNAs, additional quantitative information, such as the extent of differential expression or the proportion of regulated genes, could further improve the clarity and impact of the results.
2) The introduction is very lengthy and contains some unnecessary information, please shorten it. Also, there is a lack of transition sentences between paragraphs, which could improve the flow of the introduction and lead readers smoothly from one idea to the next.
3) Authors are requested to add ATAC-seq data analysis.
4) Authors are asked to identify DE-lncRNAs that act as miRNA target mimics
5) The authors are asked to validate DE-lncRNAs by quantitative real-time PCR analysis (qRT-PCR)

---

## Round 0.2 · accepted · Accept

I would like to thank you for accepting the referees' suggestions and improving your article based on their suggestions. Your article is ready to publish. We look forward to your next article.


·

Basic reporting

Revisions have been completed in line with the suggestions.

Experimental design

The methodology has been applied correctly.

Validity of the findings

The findings are important for other studies to be conducted in the future.

·

Basic reporting

No comment.

Experimental design

No comment.

Validity of the findings

No comment.

Additional comments

After careful reading, I think that the issues that I am concerned about are well responded.

·

Basic reporting

I have no further suggestions to make, and I believe that the authors have addressed all of my comments and that the manuscript is now ready to publish.

Experimental design

I have no further suggestions to make, and I believe that the authors have addressed all of my comments and that the manuscript is now ready to publish.

Validity of the findings

I have no further suggestions to make, and I believe that the authors have addressed all of my comments and that the manuscript is now ready to publish.

Additional comments

I have no further suggestions to make, and I believe that the authors have addressed all of my comments and that the manuscript is now ready to publish.